# Lactoferrin Supplementation during Pregnancy and Lactation Protects Adult Male Rat Offspring from Hypertension Induced by Maternal Adenine Diet

**DOI:** 10.3390/nu16162607

**Published:** 2024-08-08

**Authors:** You-Lin Tain, Chih-Yao Hou, Wei-Ling Chen, Wei-Ting Liao, Chien-Ning Hsu

**Affiliations:** 1Division of Pediatric Nephrology, Kaohsiung Chang Gung Memorial Hospital, Kaohsiung 833, Taiwan; tainyl@cgmh.org.tw (Y.-L.T.); weilingchen@cgmh.org.tw (W.-L.C.); winona0409@cgmh.org.tw (W.-T.L.); 2Institute for Translational Research in Biomedicine, Kaohsiung Chang Gung Memorial Hospital, Kaohsiung 833, Taiwan; 3College of Medicine, Chang Gung University, Taoyuan 330, Taiwan; 4Department of Seafood Science, National Kaohsiung University of Science and Technology, Kaohsiung 811, Taiwan; chihyaohou@webmail.nkmu.edu.tw; 5Department of Pharmacy, Kaohsiung Chang Gung Memorial Hospital, Kaohsiung 833, Taiwan; 6School of Pharmacy, Kaohsiung Medical University, Kaohsiung 807, Taiwan

**Keywords:** hypertension, lactoferrin, gut microbiota, nitric oxide, renin–angiotensin system, chronic kidney disease, developmental origins of health and disease (DOHaD)

## Abstract

Lactoferrin, a glycoprotein derived from breastmilk, is recognized for its health benefits in infants and children; however, its protective effects when administered during gestation and lactation against offspring hypertension remain unclear. This study aimed to investigate whether maternal lactoferrin supplementation could prevent hypertension in offspring born to mothers with chronic kidney disease (CKD), with a focus on nitric oxide (NO), renin–angiotensin system (RAS) regulation, and alterations in gut microbiota and short-chain fatty acids (SCFAs). Prior to pregnancy, female rats were subjected to a 0.5% adenine diet for 3 weeks to induce CKD. During pregnancy and lactation, pregnant rats received one of four diets: normal chow, 0.5% adenine diet, 10% lactoferrin diet, or adenine diet supplemented with lactoferrin. Male offspring were euthanized at 12 weeks of age (n = 8 per group). Supplementation with lactoferrin during gestation and lactation prevented hypertension in adult offspring induced by a maternal adenine diet. The maternal adenine diet caused a decrease in the index of NO availability, which was restored by 67% with maternal LF supplementation. Additionally, LF was related to the regulation of the RAS, as evidenced by a reduced renal expression of renin and the angiotensin II type 1 receptor. Combined maternal adenine and LF diets altered beta diversity, shifted the offspring’s gut microbiota, decreased propionate levels, and reduced the renal expression of SCFA receptors. The beneficial effects of lactoferrin are likely mediated through enhanced NO availability, rebalancing the RAS, and alterations in gut microbiota composition and SCFAs. Our findings suggest that maternal lactoferrin supplementation improves hypertension in offspring in a model of adenine-induced CKD, bringing us closer to potentially translating lactoferrin supplementation clinically for children born to mothers with CKD.

## 1. Introduction

Lactoferrin (LF), a glycoprotein from the transferrin family first isolated from cow’s milk and later identified in human milk, demonstrates a range of biological activities. These include antiviral, anti-inflammatory, antioxidant, and prebiotic properties [1,2]. Increasing evidence supports the significant clinical impact of LF on health outcomes in infants and children [3,4].

Breastmilk contains a myriad of bioactive components that play crucial roles in the health and disease risk of offspring [5,6,7]. Beyond LF, essential bioactive elements in breastmilk encompass casein and whey proteins, fatty acids, hormones, growth factors, and cytokines. These components work together synergistically to support the infant’s immune system, promote growth and development, and offer protection against infections and diseases [5,6,7].

Numerous adverse conditions during pregnancy and lactation can influence fetal development, potentially predisposing individuals to adult-onset diseases [8]. This phenomenon is recognized as the “developmental origins of health and disease” (DOHaD) [9]. Chronic kidney disease (CKD) affects 3–4% of women of reproductive age, and pregnant women with CKD are particularly vulnerable to adverse maternal and offspring outcomes [10]. Animal models utilizing an adenine diet replicate progressive kidney injury and associated complications, closely mirroring the pathophysiology observed in human CKD [11]. In one such maternal CKD model induced by adenine, adult rat offspring developed hypertension [12]. Mechanisms contributing to this phenomenon include nitric oxide (NO) deficiency, dysregulation of the renin–angiotensin system (RAS), and dysbiosis of gut microbiota.

Recent evidence underscores the beneficial role of LF in hypertension and kidney disease. In a rat model, LF acts as an antioxidant, effectively shielding against kidney oxidative injury [13]. Chronic LF administration has been shown to lower blood pressure (BP), reduce reactive oxygen species (ROS) production, and enhance antioxidant capacity in a rat model of dexamethasone-induced hypertension [14]. Further studies attribute LF’s antihypertensive effects to its regulation of NO or the RAS [15,16]. In addition to its antimicrobial properties against pathogenic bacteria, LF serves as a prebiotic that promotes the growth of selected beneficial bacteria [17]. Given the influence of gut microbiota and their metabolites, such as short-chain fatty acids (SCFAs), on BP regulation and renal physiology [18,19], LF may improve hypertension and kidney disease by restoring microbial balance.

Notably, existing research has not explored whether maternal LF supplementation can mitigate hypertension in offspring. The DOHaD theory proposes that early-life interventions can prevent adverse health outcomes by reprogramming physiological processes before disease onset [20]. Therefore, our study aimed to investigate the impact of maternal LF supplementation on hypertension in adult offspring exposed to a maternal adenine diet. We also sought to elucidate the protective mechanisms of maternal LF supplementation, particularly focusing on NO, RAS regulation, and alterations in gut microbiota.

## 2. Materials and Methods

### 2.1. Animals, Diets, and Study Design

The animal protocols utilized in this study were approved by the Institutional Animal Ethics Committee of the Kaohsiung Chang Gung Memorial Hospital (Permit # 2023061905, approval date: 16 August 2023) and adhered strictly to the ARRIVE guidelines [21]. Virgin Sprague Dawley (SD) rats (8–10 weeks old, 200–220 g) obtained from BioLASCO Co. Taipei, Taiwan, were used for mating purposes. The animals were housed in an AAALAC-accredited animal facility with daily care and maintenance provided. Throughout this study, rats had ad libitum access to water. CKD was induced by feeding the dam a 0.5% adenine diet for three weeks prior to pregnancy, as previously documented [12].

Pregnant rats were randomly allocated into four groups by utilizing a random number generator (n = 3 per group), each receiving distinct diets throughout both gestation and lactation periods (gestational day 1 to lactation day 21): the CN group received a standard AIN-93M diet (D10012M, Research Diets Inc., New Brunswick, NJ, USA); the CKD group received a standard diet supplemented with 0.5% adenine; the LF group received a standard diet supplemented with 10% LF; and the CKDLF group was fed a standard diet supplemented with 0.5% adenine plus 10% LF. Bovine LF was obtained from Bega Bionutrients (Melbourne, Australia). The LF dosage was selected based on previous rodent studies [22]. Since the average daily food intake for rats is 10 g of diet per 100 g of body weight, the corresponding dose of LF was 1 g/kg/day.

After birth, each litter was standardized to eight pups by randomly selecting four males and four females, or as close to an equal number of each sex as possible, to ensure uniform growth among the offspring. Male rats were chosen for this study due to their higher susceptibility to hypertension [23]. After weaning, the offspring were returned to a standard chow diet. The compositions of the standard chow diet and the diet supplemented with 10% LF are detailed in Table 1.

BP was recorded in offspring rats at ages of 3, 4, 8, and 12 weeks using a tail-cuff sphygmomanometer (CODA System, Kent Scientific Corporation, Torrington, CT, USA) following a published procedure [12]. Briefly, rats were acclimatized to restraint cages for 1 week prior to measurement of BP. On the day of measurement, rats were restrained, and their tails were pre-warmed for 10 to 15 min. At 12 weeks of age, a total of 32 rats were euthanized for this study. Prior to euthanasia, fresh stool samples were collected and transferred into a −80 °C freezer. The rats were anesthetized with an intraperitoneal injection of ketamine (50 mg/kg) and xylazine (10 mg/kg), and euthanized with an overdose of pentobarbital administered via intraperitoneal injection. Blood samples were collected via cardiac puncture into heparin-containing tubes. The plasma was then separated and stored at −80 °C until the analysis. Kidneys were excised, and the cortex and medulla were dissected, quickly snap-frozen, and subsequently stored in a freezer.

### 2.2. NO-Related Parameters

Parameters related to the NO pathway were measured in plasma using an Agilent 1100 HPLC system (Agilent Technologies Inc., Santa Clara, CA, USA) equipped with fluorescence detection and O-phthalaldehyde/3-mercaptopropionic acid as a derivatization agent. These NO-related parameters included L-arginine, L-citrulline, and inhibitors of nitric oxide synthase, asymmetric and symmetric dimethylarginine (ADMA and SDMA). Samples were prepared by mixing 100 μL of plasma with 4 μL of 20 μM homoarginine (Fluka, Neu Ulm, Germany) as the internal standard and 350 μL of 0.4 M borate buffer for fractionation. The mixture was then eluted with 1 mL of NH4OH/H_2_O/MEOH (10:40:50), dried under vacuum centrifugation for 5 h, and reconstituted with 60 μL of H_2_O. The recovery rate was approximately 90%. Sixty microliters of the reconstituted sample was injected into an Agilent SB-C18 column (Agilent Technologies, Inc.) at a flow rate of 1.2 mL/min. The concentrations of L-arginine, L-citrulline, ADMA, and SDMA in the standards were 1–100 μM, 1–100 μM, 0.5–5 μM, and 0.5–5 μM, respectively. To assess the availability of NO, we determined the ratio of L-arginine to ADMA [24].

### 2.3. SCFAs and Receptors

Plasma concentrations of SCFA including acetate, propionate, and butyrate were quantified using GC-MS (QP2010; Shimadzu, Kyoto, Japan) with a flame ionization detector, following established methods [25]. Separation was performed on the SGE BP GC column (21 × 0.5 µm, 30 m × 0.53 mm; Shimadzu, Kyoto, Japan). The working solutions of acetate, butyrate, and propionate used as internal and external standards were at the concentration of 10 mm and kept in a −20 °C freezer. Dry air, nitrogen, and hydrogen were supplied to the FID at 300, 20, and 30mL/min, respectively. Each sample (2 µL) was injected into the column with an inlet temperature of 200 °C, and the detector temperature was set at 240 °C. The total run time for each analysis was 17.5 min. Analytical standard grades used as internal standards for acetate and propionate were obtained from Sigma–Aldrich (St. Louis, MO, USA), while the standard for butyrate was obtained from Chem Service (West Chester, PA, USA).

Furthermore, we investigated the renal expression of SCFA receptors using quantitative PCR (qPCR). The receptors analyzed included olfactory receptor 78 (*Olfr78*), G protein-coupled receptor 109A (GPR109A, encoded by gene *Hcar2*), GPR41 (encoded by *Ffar3*), and GPR43 (encoded by *Fdar2*). RNA was extracted from rat kidney cortex tissue using TRI Reagent (Sigma, St. Louis, MO, USA) and subsequently treated with DNase I (Ambion, Austin, TX, USA) to remove any DNA contamination. Two micrograms of the RNA was then reverse-transcribed using SuperScript II RNase H-Reverse Transcriptase (Invitrogen, Bethesda, MD, USA) and random primers (Invitrogen) in a final reaction volume of 40 µL. qPCR was carried out using Quantitect SYBR Green PCR Reagents (Qiagen, Valencia, CA, USA) on a thermal cycler (iCycler, Bio-Rad, Hercules, CA, USA), following protocols described previously [25]. Ribosomal 18S RNA was utilized as the housekeeping gene. Primers were designed using GeneTool 1.0 Software (Biotools, Edmonton, AB, Canada) as we previously published [12,25]. See Table 2 for the PCR primer sequences. Gene expression levels were quantified using the comparative threshold cycle (CT) method, and all samples were assessed in duplicate.

Using qPCR, we also determined the expression of six RAS components, namely renin (encoded by gene *REN*), (pro)renin receptor (PRR, encoded by gene *Atp6ap2*), angiotensin converting enzyme-1 (ACE1, encoded by gene *Ace*), angiotensin II type 1 receptor (AT1R, encoded by gene *Agtr1a*), angiotensin converting enzyme-2 (ACE2, encoded by gene *Ace2*), and MAS receptor (encoded by gene *Mas1*).

### 2.4. 16S rRNA Gene Sequencing and Analysis

Stool DNA was extracted and sent to Biotools Co., Ltd. in New Taipei City, Taiwan, for 16S rRNA sequencing [25]. The full-length 16S genes covering V1–V9 hypervariable regions were amplified using barcoded primers. Subsequently, a multiplexed SMRTbell library (PacBio, Menlo Park, CA, USA) was prepared for sequencing. To analyze the data, a phylogenetic tree was constructed using QIIME 2’s phylogeny FastTree [26,27], illustrating the relationships among Amplicon Sequence Variants (ASVs). Alpha diversity (within-sample diversity) including Faith’s phylogenetic diversity (PD) and Shannon indices was calculated using the vegan package in R. For a beta diversity analysis, principal coordinate analysis (PCoA) plots were generated based on unweighted UniFrac distance metrics. An Analysis of Similarities (ANOSIM) was also employed to compare bacterial composition differences between groups. Furthermore, significant differential taxa between groups were identified by linear discriminant analysis (LDA) effect size (LEfSe) with an LDA score > 4 [28].

### 2.5. Statistics

All statistical analyses were performed using SPSS 17.0 (SPSS Inc., Chicago, IL, USA). The Shapiro–Wilk normality test was used to determine which data were normally distributed. Normally distributed data were presented as the mean ± the standard error of the mean. Among-group comparisons were performed using one-way ANOVA, followed by Tukey’s post hoc test. Statistical significance was set at *p* < 0.05.

## 3. Results

### 3.1. Body Weight and Blood Pressure

There were no differences in the litter size (CN = 14 ± 0.8; CKD = 13.2 ± 0.6; LF = 15.2 ± 0.8; CKDLF = 14.2 ± 0.6) and birth weights (CN = 7.7 ± 0.24 g; CKD = 7.54 ± 0.14 g; LF = 7.95 ± 0.38 g; CKDLF = 7.4 ± 0.38 g) among the four groups. Supplementing the maternal diet with adenine or LF showed no effect on pup mortality. Offspring born to dams that received LF showed significantly greater weights in the LF and CKDLF groups than those in the CN and CKD groups (Table 3). Additionally, the kidney-to-body weight ratio was highest in the LF group compared to the other three groups. Systolic blood pressures (SBPs) in offspring at different ages are depicted in Figure 1. A maternal adenine diet resulted in elevated SBP between 8 and 12 weeks of age, which was mitigated by maternal LF supplementation. Table 3 indicated that not only SBP but also diastolic blood pressure (DBP) and mean arterial pressure (MAP) were improved in the CKDLF offspring compared to the CKD offspring at 12 weeks of age.

### 3.2. NO Pathway

As summarized in Table 4, there were no differences observed in L-citrulline, L-arginine, and SDMA levels between the four groups. However, compared with the CN group, a maternal adenine diet augmented ADMA levels by approximately 44%, whereas LF supplementation decreased the adenine-induced increase by approximately 29%. Conversely, the maternal adenine diet reduced the ratio of L-arginine to ADMA (AAR), an index of NO availability, in the CKD group. This reduction was restored by 67% with maternal LF supplementation.

### 3.3. RAS

Most RAS components did not show differences between the CN and CKD groups, except for lower MAS expression in the CKD group compared with the CN group (Figure 2). Furthermore, supplementing the maternal diet with LF resulted in a decreased renal expression of renin and AT1R in the LF and CKDLF groups. However, there were no differences detected in the renal mRNA expression of PRR, ACE1, and ACE2 among the four groups.

### 3.4. Differences in Gut Microbiota Composition

Faith’s PD index (Figure 3A) and the Shannon index (Figure 3B) did not differ between the four groups. PCoA plots revealed that the microbiome compositions differed between the four groups (Figure 3C). ANOSIM of variance confirmed that there were statistically significant separations between the CN, CKD, LF, and CKDLF groups (*p* < 0.01, respectively). These results suggested that a maternal adenine or LF diet could shift the composition of the microbial community in offspring’s gut.

The LEfSe approach was utilized to discover specific taxa that were either significantly enhanced or depleted in each group, potentially serving as biomarkers. The differentially abundant taxa identified between groups are illustrated in Figure 4. In the microbiome of CN offspring, the dominant genera were *Muribaculum* and *Ruminococcus*. The CKD group exhibited a marked enrichment of *Eubacterium* and *Clostridium*. A unique feature of LF offspring was the elevated representation of *Duncaniella* and *Ligilactobacillus*. In the CKDLF group, the genus *Alistipes* was significantly more abundant.

To further explore the beneficial effects of maternal LF supplementation and delve deeper into the gut microbiota, we next investigate significant genus-level changes between the CKD and CKDLF groups. The discriminative genera identified between two groups are presented in Figure 5. Compared to the CKDLF group, the microbiome of the CKD group was dominated by *Ruminococcus, Muribaculum, Breznakia*, and *Paraeggerthella*. Conversely, genera *Anaerotignum, Fusimonas, Alistipes,* and *Duncaniella* were considerably depleted in the CKD group but abundant in the CKDLF group.

Compared with the CN group, genus *Robinsoniella* was significantly reduced in the CKD group (Figure 6). Meanwhile, this decrease was restored by maternal LF supplementation.

### 3.5. SCFAs and Receptors

Considering the involvement of SCFAs and their receptors in BP regulation, we further investigated their concentrations in plasma and SCFA receptor expression in the kidneys (Figure 7). A maternal adenine diet led to decreased levels of propionate in the CKD group compared with the CN group (Figure 7B). There were no differences in the expression of four SCFA receptors between the CN and CKD groups. However, all four receptors showed reduced expression in the CKDLF group due to a maternal combined adenine and LF diet (Figure 7C).

## 4. Discussion

In this study, we are the first to demonstrate that maternal LF supplementation can serve as a reprogramming strategy to improve hypertension in offspring within a maternal adenine diet model. The key findings reveal that (i) LF supplementation during gestation and lactation averted offspring hypertension programmed by a maternal adenine diet; (ii) maternal LF supplementation restored NO deficiency induced by the maternal adenine diet, as evidenced by increased ADMA and decreased AAR; (iii) the antihypertensive effect of LF was related to the regulation of the RAS, shown by a reduced renal expression of renin and AT1R; (iv) both maternal adenine and LF diets altered beta diversity and shifted the composition of offspring’s gut microbiota; (v) the maternal adenine diet-induced reduction in the genus *Robinsoniella* was prevented by LF supplementation; and (vi) the combined maternal adenine and LF diet led to decreased propionate levels and a reduced renal expression of SCFA receptors.

Supporting prior work showing that a maternal adenine diet leads to offspring hypertension [12], our results extend beyond prior research by demonstrating that supplementing the maternal diet with LF improves offspring hypertension in this model. While LF supplementation during pregnancy and lactation increased body weight, kidney weight, and the ratio of kidney weight-to-body weight in normal controls, whether these changes are beneficial or harmful remains to be clarified.

Consistent with the antihypertensive properties of LF reported previously [15,16], one beneficial mechanism by which CS counteracts offspring hypertension involves the regulation of the NO pathway. LF supplementation increased NO availability, characterized by decreased ADMA and increased AAR, which align with prior research supporting the effectiveness of the restoration of ADMA–NO balance in averting hypertension with developmental origins [29,30].

Another reprogramming effect of LF in preventing hypertension could be achieved through the regulation of the RAS. Growing evidence supports that aberrant RAS contributes to the developmental programming of hypertension, while RAS-based reprogramming interventions may prevent it [31]. This concept is reinforced by the present study, which demonstrated that a maternal adenine diet reduced MAS expression, while LF supplementation resulted in a decreased expression of renin and AT1R in offspring’s kidneys. In the RAS, a MAS receptor interacts with angiotensin-(1–7) in favor of vasodilatation [32]. The decrease in MAS expression due to the maternal adenine diet appears to correlate with the increase in BP observed in adult offspring. Renin initiates a cascade of events in the RAS, leading to the generation of angiotensin II, which stimulates AT1R to induce hypertension [33]. Our data suggest that the inhibition of the RAS by LF promotes vasodilation and lowers BP in adult offspring. Given that LF-derived peptides have shown antihypertensive effects by the inhibition of the RAS [34], the interplay between LF and the RAS in controlling offspring BP warrants further investigation.

Another protective effect of LF may involve its influence on gut microbiota composition. Although previous studies indicate that LF can alter gut microflora by inhibiting intestinal pathogens and acting as a prebiotic [35], little is known about the impact of maternal LF supplementation on the offspring’s microbiome. In a previous study, it was demonstrated that the relative abundance of the genus *Ruminococcus* was significantly reduced, while *Alistipes* was increased, following LF intervention [36]. Consistent with these findings, the CKD group was dominated by *Ruminococcus*, but *Alistipes* was significantly abundant in the CKDLF group. Additionally, the LEfSe analysis revealed specific bacterial taxa potentially linked to hypertension: the hypertension-depleted taxon *Ruminococcus* and the hypertension-enriched taxon *Alistipes* [37]. Furthermore, we observed that maternal CKD induced hypertension in offspring accompanied by a reduction in the genus *Robinsoniella*, an effect that was reversed with maternal LF supplementation. This observation aligns with a human study indicating a negative correlation between the abundance of *Robinsoniella* and systolic BP [38]. Accordingly, maternal LF supplementation potentially prevented hypertension in offspring by modifying these BP-associated taxa. Investigating whether these taxa could function as microbial markers for hypertension and as therapeutic targets requires further comprehensive investigation.

Supporting the prebiotic role of LF [17], our data showed that LF offspring had elevated abundance of genus *Ligilactobacillus*, known as a probiotic [39,40]. However, *Lactobacillus* and *Bifidobacterium*, well-known probiotics, were not altered by maternal LF supplementation. Whether these differences are related to direct versus reprogramming effects or variations between humans and animal models deserves further research to fully explore LF’s prebiotic abilities. We conducted a further analysis on plasma SCFAs and their receptors, given their role in BP regulation [41]. While maternal CKD led to reduced propionate levels and LF intervention decreased butyrate levels compared with the control group, there were no significant differences in SCFA levels between the CKD and CKDLF groups. Moreover, all four SCFA receptors exhibited reduced expression in the CKDLF group. Considering that each SCFA can activate multiple receptors and each receptor can respond to more than one SCFA, understanding how changes in SCFA levels and receptor expression following LF intervention contribute to its protective effects requires additional studies to untangle this complexity.

We acknowledge some limitations. Firstly, the present study did not investigate how maternal LF supplementation affected maternal and neonatal gut microbiota and metabolites. Understanding whether LF treatment during gestation and lactation can influence gut microbiota-derived SCFAs, which may contribute to BP regulation in offspring later in life, requires further investigation. Since microbial functional studies were not performed, the relative abundance of taxa may not fully reflect their biological significance. Another limitation is that our study exclusively focused on male offspring. It remains unclear whether LF supplementation exerts differential effects based on sex, underscoring the need for additional research in this area. Lastly, while our findings demonstrate the beneficial effects of LF in mitigating offspring hypertension induced by a maternal adenine diet in this specific model, it is uncertain whether these results can be directly extrapolated to other models of CKD or to human populations.

## 5. Conclusions

In conclusion, our findings suggest that maternal LF supplementation improved offspring hypertension induced by a maternal adenine diet. This improvement is likely reached through enhanced NO availability, rebalanced RAS, and alterations in gut microbiota composition. Thus, early LF supplementation shows potential for averting hypertension in offspring born to mothers with CKD.

## Figures and Tables

**Figure 1 nutrients-16-02607-f001:**
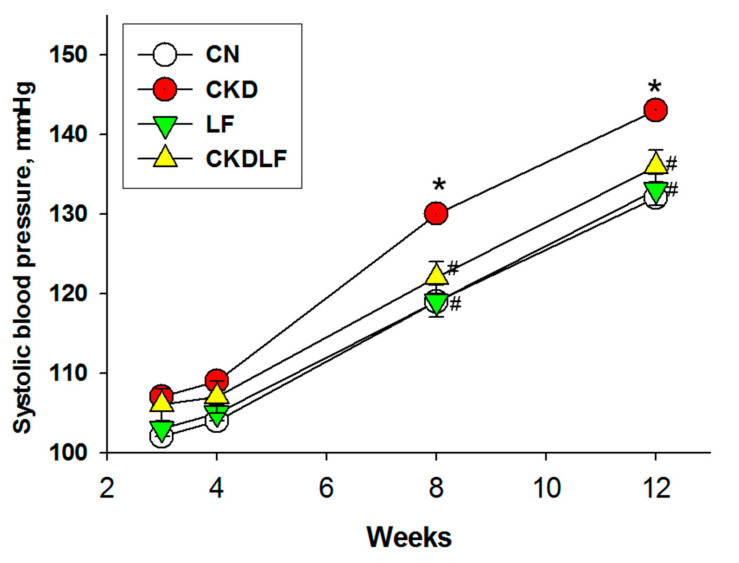
Effects of maternal adenine diet (CKD) and lactoferrin (LF) on systolic blood pressure in offspring from Week 3 to 12. N = 8/group. * *p* < 0.05 vs. CN; # *p* < 0.05 vs. CKD.

**Figure 2 nutrients-16-02607-f002:**
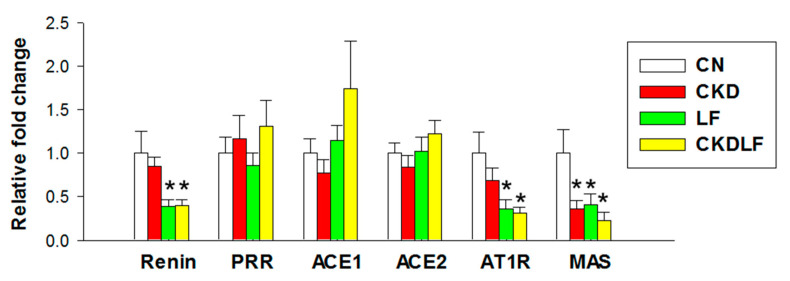
Effects of maternal adenine diet (CKD) and lactoferrin (LF) on the renin–angiotensin system at Week 12. N = 8/group. * *p* < 0.05 vs. CN.

**Figure 3 nutrients-16-02607-f003:**
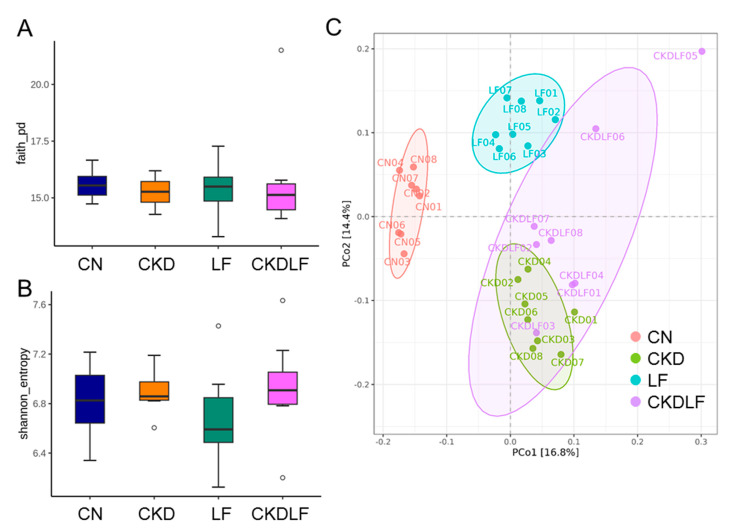
The evaluation of gut microbial biodiversity in offspring born to dams fed an adenine (CKD) or lactoferrin (LF) diet. (**A**) Faith’s phylogenic diversity (pd), (**B**) Shannon index, and (**C**) principal coordinate analysis (PCoA). Outliers are denoted by dots. Each color corresponds to a different group, with each data point representing one sample.

**Figure 4 nutrients-16-02607-f004:**
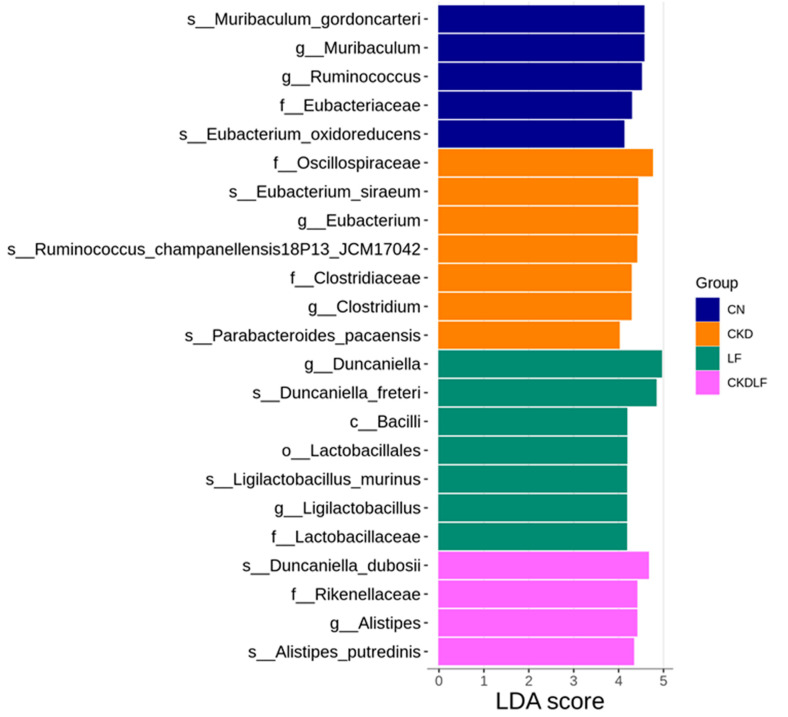
Linear discriminant analysis effect size (LEfSe) with an LDA score > 4 identified significantly differential taxa between groups. The respective group is denoted by the color of the horizontal bar.

**Figure 5 nutrients-16-02607-f005:**
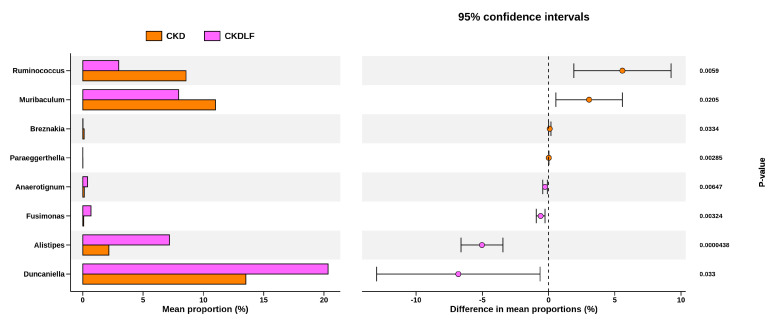
Bar plots showing the genus-level discrimination between the CKD and CKDLF groups.

**Figure 6 nutrients-16-02607-f006:**
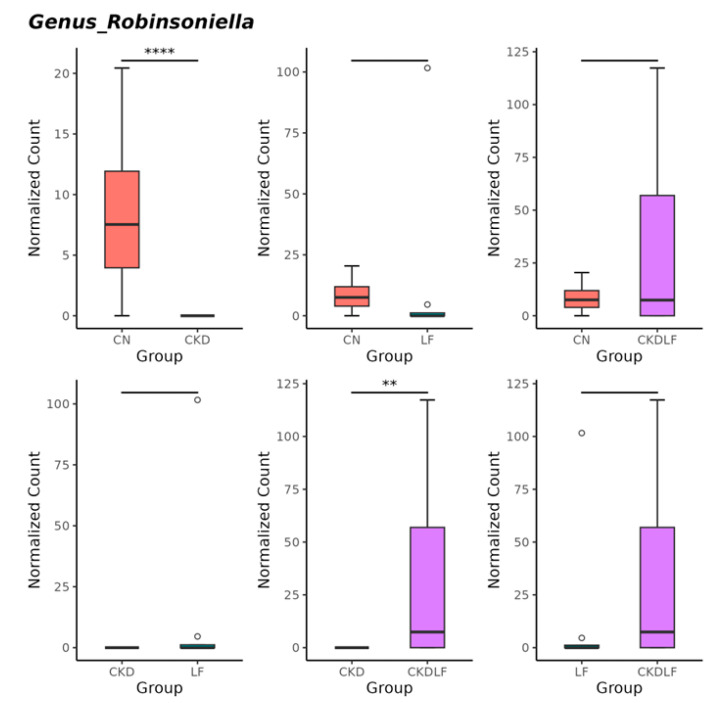
The comparison of relative abundance of genus *Robinsoniella* among the four groups. Outliers are denoted by dots. ** *p* < 0.01. **** *p* < 0.001.

**Figure 7 nutrients-16-02607-f007:**
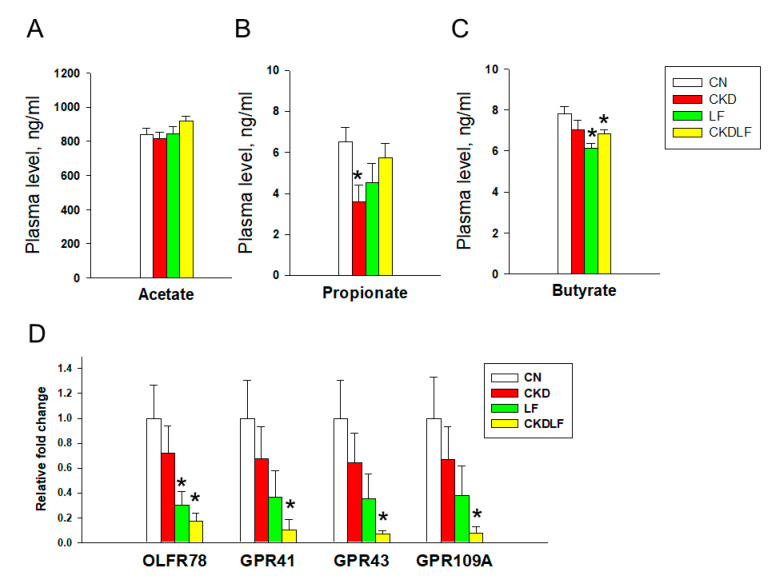
Plasma concentrations of (**A**) acetate, (**B**) propionate, and (**C**) butyrate, and (**D**) renal mRNA expression of SCFA receptors at Week 12. N = 8/group. * *p* < 0.05 vs. CN.

**Table 1 nutrients-16-02607-t001:** Composition of standard chow diet and diet supplemented with 10% LF.

	Standard Chow Diet	10% LF Diet
Nutrients		
Protein (g %)	14	14
Carbohydrate (g %)	73	73
Fat gm (g %)	4	4
Ingredients		
Casein	140	140
Cysteine	1.8	1.8
Corn starch	495.7	495.7
Maltodextrin	125	125
Sucrose	100	100
Cellulose	50	50
Soybean oil	40	40
t-butylhydroquinone	0.008	0.008
Mineral mix	35	35
Vitamin mix	10	10
Choline bitartrate	25	25
Lactoferrin	0	10
Total	1000	1010
Energy (kcal/g)	3.8	3.8

**Table 2 nutrients-16-02607-t002:** PCR primer sequences.

Gene	Accession No.	5′ Primer	3′ Primer
*Olfr78*	NM_173293.2	5 gaggaagctcacttttggtttgg 3	5 cagcttcaatgtccttgtcacag 3
*Ffar3*	NM_001108912.1	5 tcttcaccaccgtctatctcac 3	5 cacaagtcctgccaccctc 3
*Ffar2*	NM_001005877.3	5 ctgcctgggatcgtctgtg 3	5 cataccctcggccttctgg 3
*Hcar2*	NM_181476.2	5 cggtggtctactatttctcc 3	5 cccctggaatacttctgatt 3
*Ren*	J02941.1	5 aacattaccagggcaactttcact 3	5 acccccttcatggtgatctg 3
*Atp6ap2*	AB188298.1	5 gaggcagtgaccctcaacat 3	5 ccctcctcacacaacaaggt 3
*Ace1*	U03734.1	5 caccggcaaggtctgctt 3	5 cttggcatagtttcgtgaggaa 3
*Agtr1a*	NM_030985.4	5 gctgggcaacgagtttgtct 3	5 cagtccttcagctggatcttca 3
*Ace2*	NM_001012006.2	5 acccttcttacatcagccctactg 3	5 tgtccaaaacctaccccacatat 3
*Mas1*	J03823.1	5 catctctcctctcggctttgtg 3	5 cctcatccggaagcaaagg 3
*R18s*	X01117	5 gccgcggtaattccagctcca 3	5 cccgcccgctcccaagatc 3

**Table 3 nutrients-16-02607-t003:** Weights and BPs.

Groups	CN	CKD	LF	CKDLF
Body weight (BW) (g)	290 ± 12	289 ± 9	355 ± 9 *#	419 ± 9 *#
Left kidney weight (KW) (g)	1.40 ± 0.06	1.40 ± 0.04	1.85 ± 0.09 *#	1.91 ± 0.07 *#
Ratio of KW to BW (g/kg)	0.48 ± 0.01	0.48 ± 0.01	0.52 ± 0.02 *	0.46 ± 0.02 $
Systolic BP (mmHg)	132 ± 1	143 ± 1 *	133 ±1 #	136 ± 1 #
Diastolic BP (mmHg)	88 ± 3	91 ± 2	82 ± 1 #	80 ± 2 #
Mean arterial pressure (mmHg)	103 ± 2	109 ± 2 *	99 ± 1 #	100 ± 1 #

N = 8/group; * *p* < 0.05 vs. CN; # *p* < 0.05 vs. CKD; $ *p* < 0.05 vs. LF.

**Table 4 nutrients-16-02607-t004:** NO-related parameters in Plasma.

Group	CN	CKD	LF	CKDLF
L-citrulline (μM)	52.8 ± 3.3	53.5 ± 4.2	52.5 ± 4	55.9 ± 3.6
L-arginine (μM)	352.3 ± 16.5	353.9 ± 24.1	377 ± 42.1	396.6 ± 22.8
ADMA (μM)	1.81 ± 0.1	2.6 ± 0.05 *	1.76 ± 0.17 #	2.02 ± 0.19 #
SDMA (μM)	1.82 ± 0.12	1.69 ± 0.15	1.51 ± 0.08	1.65 ± 0.24
L-arginine-to-ADMA ratio (μM/μM)	200.2 ± 15.9	135.3 ± 7.2 *	225.8 ± 32.8 #	207.5 ± 19.9 #

N = 8/group; * *p* < 0.05 vs. CN; # *p* < 0.05 vs. CKD.

## Data Availability

The original data are saved at Kaohsiung Chang Memorial Hospital. Please contact the corresponding author for any inquiry.

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
