# Peer review of "Lactoferrin Supplementation during Pregnancy and Lactation Protects Adult Male Rat Offspring from Hypertension Induced by Maternal Adenine Diet"

_nutrients, 2024, doi:10.3390/nu16162607_

Round 1
Reviewer 1 Report
Comments and Suggestions for Authors
The submitted work is prepared at a very good level and its focus corresponds to your journal. I only have some comments on the material and methodology section. The methodology does not state whether the animals were randomly divided into groups. If yes, please describe how the randomization was done. It is further stated that euthanasia was performed. Please describe how this euthanasia was performed. Please describe how, that is, from which place, the blood was taken.
Author Response
RESPONSES TO REVIEWER’S COMMENTS
Reviewer #1
The submitted work is prepared at a very good level and its focus corresponds to your journal. I only have some comments on the material and methodology section.
RESPONSE: We thank Reviewer #1 for his/her generous support.
The methodology does not state whether the animals were randomly divided into groups. If yes, please describe how the randomization was done.
RESPONSE: Per the suggestion, we have revised our statement as follows.
Lines 91-92: “Pregnant rats were randomly allocated into four groups by utilizing a random number generator….”
It is further stated that euthanasia was performed. Please describe how this euthanasia was performed. Please describe how, that is, from which place, the blood was taken.
RESPONSE: In response to the suggestion, we have included the following statements in the Methods section to clarify the procedure.
Lines 115-119: “The rats were anesthetized with an intraperitoneal injection of ketamine (50 mg/kg) and xylazine (10 mg/kg), and euthanized with an overdose of pentobarbital administered via intraperitoneal injection. Blood samples were collected via cardiac puncture into heparin-containing tubes. The plasma was then separated and stored at −80°C until analysis.”

Reviewer 2 Report
Comments and Suggestions for Authors
Thank you for the opportunity to review the manuscript entitled "Lactoferrin Supplementation During Pregnancy and Lactation Protects Adult Male Rat Offspring from Hypertension Induced by Maternal Adenine Diet". My comments on this manuscript are included below.
In line 34, the authors mention that "Breastmilk contains a myriad of bioactive components". However, the number of active ingredients in milk is more or less known. Therefore, more precise data can be cited in this sentence.
In line 36, the authors mention that "Lactoferrin ... (was) ... originally isolated from breastmilk". However, lactoferrin was originally isolated from cow's milk, and its presence in human milk was discovered later.
It is worth considering a change in the organization of the introduction. The introduction can start from paragraph 2 and then move on to the description of the role of lactoferrin by combining paragraphs 1 and 3.
In line 72, please add a citation to the ARRIVE guidelines.
Please provide the age and weight of the rats obtained for the study.
The small size of the study groups significantly limits the reliability of the obtained results.
Please provide the composition/nutritional value of the diet given to the rats during this study.
Please indicate from which day of pregnancy the intervention period started.
Please provide the number of animals born to each pregnant rat and the average weight gain during pregnancy. Please provide birth weight for each animal.
Please state until what day after delivery the intervention was continued.
How exactly was the standardization of the size of the study groups carried out?
Were any adverse effects observed during the intervention?
In Materials and Methods section please specify when the blood pressure measurements were conducted. Were measurements also taken during pregnancy and lactation?
The description method of NO-related parameters, SCFA, and receptors should be presented more accurately.
Gene names should be italicized.
Was the lactoferrin used in the study of bovine origin? What was the average daily dose of lactoferrin consumed by the animals? What was the feed dosage given to the animals during the intervention period?
Please specify which test was used to check the normality of the data distribution.
The data presented in Figure 6 should be shown in a table instead.
The Author Contribution Statement should be prepared according to the CRediT taxonomy.
In my opinion, the Data Availability Statement needs revision. In this section, the authors should indicate where the raw data from this study is available and how to access it.
Author Response
Reviewer #2
Thank you for the opportunity to review the manuscript entitled "Lactoferrin Supplementation During Pregnancy and Lactation Protects Adult Male Rat Offspring from Hypertension Induced by Maternal Adenine Diet". My comments on this manuscript are included below.
RESPONSE: We thank the reviewer #2 for the efforts and the constructive comments on the work.
In line 34, the authors mention that "Breastmilk contains a myriad of bioactive components". However, the number of active ingredients in milk is more or less known. Therefore, more precise data can be cited in this sentence.
RESPONSE: As suggested, we have added the following statements and references for a more detailed explanation.
Lines 45-50: “Breastmilk contains a myriad of bioactive components that play crucial roles in the health and disease risk of offspring [5-7]. Beyond LF, essential bioactive elements in breastmilk encompass casein and whey proteins, fatty acids, hormones, growth factors, and cytokines. These components work together synergistically to support the infant’s immune system, promote growth and development, and offer protection against infections and diseases [5-7].”
In line 36, the authors mention that "Lactoferrin ... (was) ... originally isolated from breastmilk". However, lactoferrin was originally isolated from cow's milk, and its presence in human milk was discovered later.
RESPONSE: We have made the necessary corrections accordingly.
Lines 40-41: “Lactoferrin, a glycoprotein from the transferrin family first isolated from cow's milk and later identified in human milk, demonstrates a range of biological activities.”
It is worth considering a change in the organization of the introduction. The introduction can start from paragraph 2 and then move on to the description of the role of lactoferrin by combining paragraphs 1 and 3.
RESPONSE: As suggested, we have reordered the paragraphs to enhance readability.
In line 72, please add a citation to the ARRIVE guidelines.
RESPONSE: We have provided the reference accordingly.
- Percie du Sert, N.; Hurst, V.; Ahluwalia, A.; Alam, S.; Avey, M.T.; Baker, M.; Browne, W.J.; Clark, A.; Cuthill, I.C.; Dirnagl, U.; et al. The ARRIVE guidelines 2.0: Updated guidelines for reporting animal research. PLoS Biol. 2020, 18, e3000410.
Please provide the age and weight of the rats obtained for the study.
RESPONSE: Per suggestion, we have provided the age and weight in Materials and Methods section.
Lines 85-86: “Virgin Sprague Dawley (SD) rats (8–10 weeks old, 200–220 g) obtained from BioLASCO Co. Taipei, Taiwan, were used for mating purposes.”
The small size of the study groups significantly limits the reliability of the obtained results.
RESPONSE: In adherence to the 3Rs principle, we strive to reduce animal use by measuring multiple variables in each rat. Blood pressure (BP) is a primary variable in these experiments. According to our previous research, a power analysis using an alpha level of 0.05 and 80% power determined that a minimum sample size of n = 8 per group was necessary to achieve statistical significance. We hope the Reviewer understands our rationale for this approach.
Please provide the composition/nutritional value of the diet given to the rats during this study.
RESPONSE: As suggested, we have provided the composition of diet in Table 1.
Please indicate from which day of pregnancy the intervention period started.
RESPONSE: As suggested, we have rephrased the following sentence for clarity.
Lines 91-93: “Pregnant rats were randomly allocated into four groups by utilizing a random number generator (n=3 per group), each receiving distinct diets throughout both gestation and lactation periods (gestational day 1 to lactation day 21)”
Please provide the number of animals born to each pregnant rat and the average weight gain during pregnancy. Please provide birth weight for each animal.
RESPONSE: Since our primary focus is on offspring outcomes, we did not measure the dam’s weight gain. As requested, we have provided the litter size and birth weights as follows.
Lines 196-198: “There were no differences in the litter size (CN = 14 ± 0.8; CKD = 13.2 ± 0.6; LF = 15.2 ± 0.8; CKDLF = 14.2 ± 0.6) and birth weights (CN = 7.7 ± 0.24 g; CKD = 7.54 ± 0.14 g; LF = 7.95 ± 0.38 g; CKDLF = 7.4 ± 0.38 g) among the four groups.”
Please state until what day after delivery the intervention was continued.
RESPONSE: As suggested, we have rephrased the following sentence for clarity.
Lines 91-93: “Pregnant rats were randomly allocated into four groups by utilizing a random number generator (n=3 per group), each receiving distinct diets throughout both gestation and lactation periods (gestational day 1 to lactation day 21)”
How exactly was the standardization of the size of the study groups carried out?
RESPONSE: Per the suggested, we have rephrased the following sentence for clarity.
Lines 101-103: “After birth, each litter was standardized to eight pups by randomly selecting four males and four females, or as close to an equal number of each sex as possible, to ensure uniform growth among the offspring.”
Were any adverse effects observed during the intervention?
RESPONSE: We did not observe any major adverse effects from LF supplementation. Additionally, adding adenine or LF to the maternal diet had no impact on pup mortality.
In Materials and Methods section please specify when the blood pressure measurements were conducted. Were measurements also taken during pregnancy and lactation?
RESPONSE: We have included the age of the rats when they received BP measurements. However, BP measurements were not taken in the dams for this study.
Lines 109-111: “BP was recorded at age of 3, 4, 8, and 12 weeks using a tail-cuff sphygmomanometer (CODA System, Kent Scientific Corporation, Torrington, CT, USA) following our published procedure [10].”
The description method of NO-related parameters, SCFA, and receptors should be presented more accurately.
RESPONSE: As suggested, we have provided a more detailed rephrasing of the methodology for NO, SCFA, and receptors.
Lines 127-135: “Samples were prepared by mixing 100 μL of plasma with 4 μL of 20 μM homoarginine (Fluka, Neu Ulm, Germany) as the internal standard and 350 μL of 0.4 M borate buffer for fractionation. The mixture was then eluted with 1 mL of NH4OH/H2O/MEOH (10:40:50), dried under vacuum centrifugation for 5 hours, and reconstituted with 60 μL of H2O. The recovery rate was approximately 90%. Sixty microliters of the reconstituted sample was injected into an Agilent SB-C18 column (Agilent Technologies, Inc) at a flow rate of 1.2 mL/min. The concentrations of L-arginine, L-citrulline, ADMA, and SDMA in the stand-ards were 1–100 μM, 1–100 μM, 0.5–5 μM, and 0.5–5 μM, respectively.”
Lines 141-150: “Separation was performed on the SGE BP GC column (21×0.5 µm, 30 m×0.53 mm; Shimadzu, Kyoto, Japan). The working solutions of acetate, butyrate, and propionate used as internal and external standards were at the concentration of 10 mm and kept at−20 °C freezer. Dry air, nitrogen, and hydrogen were supplied to the FID at 300, 20, and 30mL/min, respectively. Each sample (2 µL) was injected into the column with an inlet temperature of 200°C, and the detector temperature was set at 240°C. The total run time for each analysis was 17.5 minutes. Analytical standard grades used as internal standards for acetate and propionate were obtained from Sigma–Aldrich (St. Louis, MO, USA), while the standard for butyrate was obtained from Chem Service (West Chester, PA, USA).”
Lines 152-165: “The receptors analyzed included olfactory receptor 78 (Olfr78), G protein-coupled receptor 109A (GPR109A, encoded by gene Hcar2), GPR41 (encoded by Ffar3), and GPR43 (en-coded by Fdar2). RNA was extracted from rat kidney cortex tissue using TRI Reagent (Sigma, St. Louis, MO) and subsequently treated with DNase I (Ambion, Austin, TX) to remove any DNA contamination. Two micrograms of the RNA were then re-verse-transcribed using SuperScript II RNase H-Reverse Transcriptase (Invitrogen, Be-thesda, MD) and random primers (Invitrogen) in a final reaction volume of 40 µL. qPCR was carried out using Quantitect SYBR Green PCR Reagents (Qiagen, Valencia, CA) on a thermal cycler (iCycler, Bio-Rad, Hercules, CA, USA), following protocols described previously [25]. Ribosomal 18S RNA was utilized as the housekeeping gene. Primers were designed using GeneTool Software (Biotools, Edmonton, Alberta, Canada) as we previously published [12,25]. See Table 2 for the PCR primer sequences. Gene expression levels were quantified using the comparative threshold cycle (CT) method, and all samples were assessed in duplicate.”
Gene names should be italicized.
RESPONSE: We have corrected the gene names and formatted them in italics.
Was the lactoferrin used in the study of bovine origin? What was the average daily dose of lactoferrin consumed by the animals? What was the feed dosage given to the animals during the intervention period?
RESPONSE: Yes, we have included information on bovine lactoferrin in the Materials and Methods section. Additionally, we provided the expected LF dose during intervention period as follows.
Lines 97-100:” Bovine LF was obtained from Bega Bionutrients (Melbourne, Australia). The LF dosage was selected based on previous rodent studies [22]. Since the average daily food intake for rats is 10 g of diet per 100 g of body weight, the corresponding dose of LF was 1 g/kg/day.”
Please specify which test was used to check the normality of the data distribution.
RESPONSE: Per the request, we have provided the test in 2.5. Statistics section.
Lines 190-192: “The Shapiro–Wilk normality test was used to determine which data were normally distributed. Normally distributed data were presented as mean ± the standard error of the mean.”
The data presented in Figure 6 should be shown in a table instead.
RESPONSE: Since the abundance shown in the figure provides a clearer comparison than the table, we sincerely hope the reviewer will agree to retain it in its current form.
The Author Contribution Statement should be prepared according to the CRediT taxonomy.
RESPONSE: Corrections have been implemented.
In my opinion, the Data Availability Statement needs revision. In this section, the authors should indicate where the raw data from this study is available and how to access it.
RESPONSE: As suggested, we have rephrased our statements as follows.
“The original data are saved at Kaohsiung Chang Memorial Hospital. Please contact the corresponding author for any inquiry.”

Reviewer 3 Report
Comments and Suggestions for Authors
The article presented by the authors is interesting. Breastfeeding is of great importance in the development of the newborn. I have a doubt that arises with this type of work. If I am right, rat milk does not have lactoferrin, then the offspring are given a molecule that is not really present in their mother's milk. Breast milk does have lactoferrin. Therefore, the data obtained in rats can really be extrapolated ? On the other hand, it is not very well explained how the hypertensive diet in the mother ends up giving rise to hypertension in her offspring, and this is an important point of the work. Then there is a point that is not clear to me in the work. We have four groups with 3 rats per group. After giving birth, it is normalized to 8 offspring per rat? So per group we would have 32 rats, right? But for example, in the footer of Table 2 it says N=8/group and in figure 3E there are also only 8 per group. The authors have to clarify this point.
The study is not correctly reflected in the abstract. It is not clearly indicated which parameters are evaluated (There is no mention of microbiota, SCFA, receptor analysis). This is important to indicate so that the reader has context of the work. Likewise, the results section is too generic and no numerical results are indicated for anything.
In the material and methods there is no information on what lactoferrin was used or where it was obtained. Is it cow lactoferrin?
During the three months of the study, did the rat pups only receive breast milk?
Line 58: Beneficial bacteria instead of probiotics, this last term implies more things.
Line 70: Include the name of the institution.
Line 87: I am not clear if the analyses are in the mothers or in the offspring.
Line 88: Avoid the use of words like our.
Line 97: The authors have to correctly explain the method used. In this case it is not, the analysis parameters, sample treatment are missing….
Line 109: Any previous treatment of the sample?
Line 113: What kit or reagent was used to extract the RNA? How much sample was used?
Line 117: Were the primers designed in this study? Are they primers from a previous study?
Table 1: Name of genes in italics?
Line 184: Avoid using words like dramatically. Also, there is really no such dramatic difference between the groups of interest, which are CKD and CKDLF.
Figure 6: The number of readings of the Robinsoniella genus is very small in many of the comparisons indicated here. To perform the analyses, what minimum number of readings was established as the cut-off point? This is important because sometimes with a very small number of readings, the differences may be more due to a technical issue than to biological significance. Therefore, it is very important that the authors review this point.
Line 218: The expression of the receptors?
Did the authors not consider measuring the levels of SCFA in the feces? Since one of the hypotheses is that lactoferrin has a probiotic effect, it would be interesting to see the changes that occur in the feces with different types of diet.
Author Response
Reviewer #3
The article presented by the authors is interesting. Breastfeeding is of great importance in the development of the newborn. I have a doubt that arises with this type of work. If I am right, rat milk does not have lactoferrin, then the offspring are given a molecule that is not really present in their mother's milk. Breast milk does have lactoferrin. Therefore, the data obtained in rats can really be extrapolated ? On the other hand, it is not very well explained how the hypertensive diet in the mother ends up giving rise to hypertension in her offspring, and this is an important point of the work.
RESPONSE: We thank Reviewer #3 for the efforts and the constructive comments on our work. We understand that lactoferrin is not present in rat breastmilk. However, since bovine lactoferrin has been widely utilized as a functional feed supplement in clinical practice, we tested whether its use during pregnancy and lactation can prevent hypertension in adult offspring exposed to a maternal adenine diet. As our previous studies have shown that a maternal adenine diet can induce hypertension in offspring, we primarily focused on the protective role of lactoferrin rather than the effects of the adenine diet itself. We hope the Reviewer understands our rationale for this approach.
Then there is a point that is not clear to me in the work. We have four groups with 3 rats per group. After giving birth, it is normalized to 8 offspring per rat? So per group we would have 32 rats, right? But for example, in the footer of Table 2 it says N=8/group and in figure 3E there are also only 8 per group. The authors have to clarify this point.
RESPONSE: Sorry for the misunderstanding. We have added the following statements to describe the litter standardization for clarity. After giving birth, it is normalized to 8 offspring (4 males and 4 females) per litter. Although we had more than 8 male rat offspring per group, we used the same 8 male offspring per group for every experiment.
Lines 101-103: “After birth, each litter was standardized to eight pups by randomly selecting four males and four females, or as close to an equal number of each sex as possible, to ensure uniform growth among the offspring.”
The study is not correctly reflected in the abstract. It is not clearly indicated which parameters are evaluated (There is no mention of microbiota, SCFA, receptor analysis). This is important to indicate so that the reader has context of the work. Likewise, the results section is too generic and no numerical results are indicated for anything.
RESPONSE: We did not present parameters in details as the abstract limits 200 words maximum. As requested, we have added the following statements for clarity.
“This study aimed to investigate whether maternal lactoferrin supplementation could prevent hypertension in offspring born to mothers with chronic kidney disease (CKD), with a focus on nitric oxide (NO), renin-angiotensin system (RAS) regulation, and alterations in gut microbiota and short-chain fatty acids (SCFAs).”
“The maternal adenine diet caused a decrease in the index of NO availability, which was restored by 67% with maternal LF supplementation. Additionally, LF was related to the regulation of the RAS, as evidenced by reduced renal expression of renin and the angiotensin II type 1 receptor. Combined maternal adenine and LF diets altered beta diversity, shifted the offspring’s gut microbiota, decreased propionate levels, and reduced renal expression of SCFA receptors. The beneficial effects of lactoferrin are likely mediated through enhanced NO availability, rebalancing the RAS, and alterations in gut microbiota composition and SCFAs.”
In the material and methods there is no information on what lactoferrin was used or where it was obtained. Is it cow lactoferrin?
RESPONSE: As requested, we have included information on bovine lactoferrin in the Materials and Methods section.
Lines 97-98: “Bovine LF was obtained from Bega Bionutrients (Melbourne, Australia).”
During the three months of the study, did the rat pups only receive breast milk?
RESPONSE: Rat pups received breast milk during lactation for 3 weeks. After weaning, the offspring were returned to a standard chow diet (Line 104).
Line 58: Beneficial bacteria instead of probiotics, this last term implies more things.
RESPONSE: The correction has been made accordingly.
Line 70: Include the name of the institution.
RESPONSE: The correction has been made accordingly.
Line 87: I am not clear if the analyses are in the mothers or in the offspring.
RESPONSE: We have rephrased this sentence for clarity.
Lines 109-111: “BP was recorded in offspring rats at age of 3, 4, 8, and 12 weeks using a tail-cuff sphygmomanometer (CODA System, Kent Scientific Corporation, Torrington, CT, USA) following our published procedure [12].”
Line 88: Avoid the use of words like our.
RESPONSE: As suggested, we have reviewed the entire text to eliminate the use of the word "our."
Line 97: The authors have to correctly explain the method used. In this case it is not, the analysis parameters, sample treatment are missing….
RESPONSE: As requested, we have provided a more detailed rephrasing of the methodology for NO and other methods.
Lines 127-135: “Samples were prepared by mixing 100 μL of plasma with 4 μL of 20 μM homoarginine (Fluka, Neu Ulm, Germany) as the internal standard and 350 μL of 0.4 M borate buffer for fractionation. The mixture was then eluted with 1 mL of NH4OH/H2O/MEOH (10:40:50), dried under vacuum centrifugation for 5 hours, and reconstituted with 60 μL of H2O. The recovery rate was approximately 90%. Sixty microliters of the reconstituted sample was injected into an Agilent SB-C18 column (Agilent Technologies, Inc) at a flow rate of 1.2 mL/min. The concentrations of L-arginine, L-citrulline, ADMA, and SDMA in the stand-ards were 1–100 μM, 1–100 μM, 0.5–5 μM, and 0.5–5 μM, respectively.”
Line 109: Any previous treatment of the sample?
RESPONSE: We have provided a more detailed rephrasing of the methodology for SCFAs.
Lines 141-150: “Separation was performed on the SGE BP GC column (21×0.5 µm, 30 m×0.53 mm; Shimadzu, Kyoto, Japan). The working solutions of acetate, butyrate, and propionate used as internal and external standards were at the concentration of 10 mm and kept at−20 °C freezer. Dry air, nitrogen, and hydrogen were supplied to the FID at 300, 20, and 30mL/min, respectively. Each sample (2 µL) was injected into the column with an inlet temperature of 200°C, and the detector temperature was set at 240°C. The total run time for each analysis was 17.5 minutes. Analytical standard grades used as internal standards for acetate and propionate were obtained from Sigma–Aldrich (St. Louis, MO, USA), while the standard for butyrate was obtained from Chem Service (West Chester, PA, USA).”
Line 113: What kit or reagent was used to extract the RNA? How much sample was used?
RESPONSE: Per the suggestion, we have rephrased the following sentence for clarity.
Lines 154-158: “RNA was extracted from rat kidney cortex tissue using TRI Reagent (Sigma, St. Louis, MO) and subsequently treated with DNase I (Ambion, Austin, TX) to remove any DNA contamination. Two micrograms of the RNA were then reverse-transcribed using SuperScript II RNase H-Reverse Transcriptase (Invitrogen, Bethesda, MD) and random primers (Invitrogen) in a final reaction volume of 40 µL.”
Line 117: Were the primers designed in this study? Are they primers from a previous study?
RESPONSE: Per the suggestion, we have added the following statements for clarity.
Lines 161-163: “Primers were designed using GeneTool Software (Biotools, Edmonton, Alberta, Canada) as we previously published [12,25].”
Table 1: Name of genes in italics?
RESPONSE: We have corrected the gene names and formatted them in italics.
Line 184: Avoid using words like dramatically. Also, there is really no such dramatic difference between the groups of interest, which are CKD and CKDLF.
RESPONSE: Based on your suggestion, we have removed the term "dramatically."
Figure 6: The number of readings of the Robinsoniella genus is very small in many of the comparisons indicated here. To perform the analyses, what minimum number of readings was established as the cut-off point? This is important because sometimes with a very small number of readings, the differences may be more due to a technical issue than to biological significance. Therefore, it is very important that the authors review this point.
RESPONSE: We did not set a minimum number as a cut-off point in the metabolomic analysis. Although the abundance of the Robinsoniella genus is not in the top 10%, it constitutes around 50% of the total communities. Since microbial functional studies were not performed, the relative abundance of taxa may not fully reflect their biological significance. We agree with the Reviewer and have added this issue as a limitation in the Discussion:
Lines 354-355: "Since microbial functional studies were not performed, the relative abundance of taxa may not fully reflect their biological significance."
Line 218: The expression of the receptors?
RESPONSE: The correction has been made accordingly.
Did the authors not consider measuring the levels of SCFA in the feces? Since one of the hypotheses is that lactoferrin has a probiotic effect, it would be interesting to see the changes that occur in the feces with different types of diet.
RESPONSE: Our prior work measured plasma and fecal SCFAs simultaneously and observed that they are not identical. Plasma levels of SCFAs appear to be more related to clinical phenotypes, which is why we primarily focused on plasma SCFA levels in the present study. Additionally, we did not investigate how maternal LF supplementation affected maternal and neonatal gut microbiota, which we acknowledge as a limitation. Measuring fecal SCFAs in dams and neonatal offspring should be the target to address this question. Nevertheless, we agree with the Reviewer on this comment and will pursue further studies in this aspect.

Round 2
Reviewer 2 Report
Comments and Suggestions for Authors
The authors have correctly addressed my comments.
Reviewer 3 Report
Comments and Suggestions for Authors
The authors have responsed to all my concerns.